# The Use of Certainty in COVID-19 Reporting in Two Austrian Newspapers

**Johannes Scherling *** and **Anouschka Foltz**

Institute of English Studies, University of Graz, Heinrichstraße 36/2, 8010 Graz, Austria;
anouschka.foltz@uni-graz.at
* Correspondence: johannes.scherling@uni-graz.at

**Abstract:** Over the course of the COVID-19 pandemic, in many parts of the Global North, the public has looked to the media as an important source of information about new developments and measures to combat the spread of the virus. The main measure propagated by governments in this respect was the mass vaccination program. In this context, two important concepts in the media coverage were herd immunity and vaccine efficacy, both of which had to be reevaluated over time. In this study, we looked at the discursive construction of "the science" in the discourse on herd immunity and vaccine efficacy in two Austrian broadsheet newspapers. Our corpus-based analysis showed a tendency to overuse linguistic items implying certainty in the face of a very fast-changing, and thus uncertain, situation. We also found evidence that these two Austrian media outlets no longer function as corrective of power, but have taken on the role of mediators of sanctioned government narratives. We argue that the uncritical reporting of government narratives in such a fluid situation has led to unresolved and unreflected inconsistencies in the reporting, arguably decreasing the public's trust in the accuracy of the COVID-19 information presented in the media.

**Keywords:** COVID-19; herd immunity; vaccine efficacy; media; Austria; science; certainty

## 1. Introduction

"The first casualty when war comes, is truth" is a quote attributed to U.S. Senator Hiram Johnson during World War I (Knightley 2003, p. vii) and referred to the reliance, in times of conflict, on information from the government, "which is not [...] going to display information and argue publicly against what it wishes to do" (Bagdikian 2004, p. 84). While the original quote was made in the context of armed conflict, it can be argued to be similar in any kind of perceived or actual emergency, when the public looks to their government and leaders to provide them with outlooks and solutions. The—still ongoing—medical crisis surrounding COVID-19 can be seen as a case in point. The words that were used by governments and, subsequently, by the media to describe the crisis were metaphors of war, where the virus was the invisible enemy that had to be fought in a collective effort through social distancing, lockdowns, and other safety measures; people, conversely, were afraid and cautious like in an actual conflict situation. Information, at least in many parts of the Global North, was often centralized through the government and distributed by media outlets that had come to see themselves as government messengers, rather than as government scrutinizers. In an editorial in the Austrian newspaper *Kleine Zeitung* on 22 March 2020, at the onset of the COVID-19 pandemic, the Editor in Chief wrote, "The media, too, are now not a corrective, but mediators" (Patterer 2020), ushering in an era in which the media openly renounced their function of holding power to account and, under the guise of a medical emergency, opened the door to the danger of giving exclusive definitional power to those *in* power. In times of uncertainty—such as a global pandemic—"policy makers and health experts sometimes shy away from communicating scientific uncertainty, fearing that the uncertainty will generate mistrust" (Wegwarth et al. 2020, p. 1). It thus stands to reason

that newspapers that no longer see themselves as a corrective are likely to uncritically echo such official notions of certainty instead of scrutinizing them. In the current paper, we explored how—in times of such high reliance on official sources and narratives—two Austrian media outlets dealt with information about aspects of COVID-19 that steadily evolved over the course of the pandemic (as new variants emerged that differed in levels of severity and infectiousness; cf., Lippi and Henry 2021; Ramesh et al. 2021), focusing on herd immunity and vaccine efficacy. We analyzed to what extent the messaging from such official sources, which was characterized by certainty and definite claims, was reflected, scrutinized, and reported on by two Austrian newspapers.

## 1.1. COVID-19 and the Vaccination Program

COVID-19, also called SARS-CoV-2, is a respiratory virus that was first detected in the city of Wuhan in China in late 2019 (WHO 2020a). Due to its rapid spread beyond the borders of China, it was declared a pandemic in March 2020 (WHO 2020b). From very early in the pandemic, expectations were raised for a vaccine against the Coronavirus. Eventually, four vaccine candidates were rolled out under emergency authorization (in the U.S.) and conditional marketing authorization (in the EU), and a global vaccination program was launched, based on the purpose of achieving herd immunity, an epidemiological concept that denotes "the indirect protection from an infectious disease that happens when a population is immune either through vaccination or immunity developed through previous infection" (WHO 2020c). On a global level, the WHO declared it their goal to vaccinate at least 70% of the world's population to achieve the following purposes (WHO 2021, p. 3):

1. To reduce mortality and severe morbidity and hospitalization;
2. To resume most socio-economic activities;
3. To reduce transmission and future risks.

These goals were then also proclaimed on a European level as (1) the reduction of pressure on the healthcare system, (2) the reduction of overall COVID-19 severity and mortality, (3) the re-opening of society, and (4) disease elimination (ECDC 2021, p. 2). In the Austrian context, which this study drew on, the ministry of health issued a vaccination plan according to which a vaccination rate, first of 50%+ (Kurier 2020a), then of 60–65% (Kleine Zeitung 2021), and by the end of 2021 of 90% was desirable (Parlament 2021). Achieving herd immunity by means of vaccination became one of the major goalposts of the pandemic response.

## 1.2. Media, Sources, and the Use of Certainty

Communicating these goals and how to achieve them became the tasks of the news media (cf., Patterer 2020; Grimberg 2020). Media scholars (Cook [1998] 2005; Davies 2008; Herman and Chomsky [1988] 2002) have long argued that official sources (such as government officials), as well as officially approved experts (e.g., medical or political analysts), are the media's primary and preferred source—primarily because their information is perceived as authoritative, "recognizable and credible by their status and prestige" (Herman and Chomsky [1988] 2002, p. 19) and, thus, does not need vetting (Cook [1998] 2005). This, however, also implies a certain dependency that might be counterproductive to the media's role as the fourth estate, i.e., to not only pass on information, but also critically evaluate it (cf., Ghersetti et al. 2023). In what could be construed as a Freudian slip, BBC's Fran Unsworth, in John Pilger's documentary *The War You Don't See*, said of government spokespeople: "they are entitled to their opinions and we have a duty to report it" (Pilger 2010, 08:54). In an ideal world, of course, journalists, being the fourth power in democratic societies (Gentzkow et al. 2006), would wish to question what officials say rather than merely report it. Bagdikian (2004, p. 19) maintains that "major news media overwhelmingly quote the men and women who lead hierarchies of power" who "seldom wish to publicize information that discloses their mistakes or issues they wish to keep in the background or with which they disagree," adding that "[o]fficials do not always say the whole truth" (ibid.). It becomes clear how this can be of import in the case of a crisis where mistakes

may have grave consequences that should be revealed by investigative journalism, in the absence of which mistakes will be continued or covered up. In other words, in the case of a pandemic, which by definition is a health crisis, journalists, therefore, have a heightened responsibility to hold those in power to account.

When media afford particular sources the right to speak, they give them the power and privilege to frame the narrative. This is particularly the case if their voices are in the majority and alternative voices are rarely heard, but also applies when journalists "go beyond simply reporting a view and directly endorse it" (Philo and Berry 2011, p. 176) such as by using a reporting verb such as "state" instead of "claim" (e.g., "She stated the situation was dramatic."), or even no reporting verb at all ("The situation was dramatic"). By repeatedly quoting the same officials and experts, the news media first confer, then reaffirm the officials' and experts' nature as authoritative voices (Cook [1998] 2005, p. 92), and by selecting experts that mostly echo similar or identical viewpoints, this results in the illusion of "the one truth" and of universal consensus. Expert, in this context, is an honorary title, as it were, that is given not solely or even primarily on the basis of expertise and inside knowledge, but more so with regard to alignment with official positions or official policy goals (Herman and Chomsky [1988] 2002, p. 24; Hollar 2022). When this is conflated with an uncritical echo chamber for official narratives or sources in the media, it stands to reason that the claims of certainty from above will be mirrored, and not qualified, by the reporting journalists.

When at the same time, alternative and more government-independent expert voices and narratives are pushed back, this may prove even more problematic, especially in a crisis where different takes of the situation are bound to arise. As Bagdikian states, "[i]ndependent documented information is most needed at the time when officialdom announces a crucial decision. That is when the audience is paying full and anxious attention to conflicting views being debated" (Bagdikian 2004, p. 82). In a time of crisis, therefore, getting information from a variety of sources, including independent sources without any vested interests in an impending or unfolding crisis, is vital. It was one of the goals of this paper to investigate to what extent biased official views may have been naturalized into unquestionable scientific facts by using linguistic devices of certainty, drawing on data from two Austrian media outlets.

### 1.3. Conception of Science

To make sense of scientific information, "we must look at the way in which scientific enquiry is conducted" (Carey 2011, p. 2). Most importantly, the scientific method involves choices—and these choices, so-called researcher degrees of freedom, can directly affect research outcomes. Scientists need to decide how to design their studies, how and from whom to collect (or exclude) data, and how to code, analyze, and interpret the data. All of these steps involve numerous, often arbitrary, choices that can bias scientific studies (Wicherts et al. 2016). Researchers, often inadvertently and with no malicious intent, may make numerous choices that increase the likelihood of obtaining statistically significant results because scientific journals are more likely to publish interesting and statistically significant results (Koletsi et al. 2009). If research is funded by an organization, researchers may make choices that increase the likelihood of obtaining results that the funder would consider desirable. For any piece of research, we should therefore ask ourselves: Who did the study? Which research tradition do they come from? Who funded the study? What outcomes might the funder consider beneficial? What choices did the researchers make? Were these reasonable choices? What alternative choices could the researchers have made? How might the researchers' choices have affected the results? How did the researchers interpret the data? Were these reasonable interpretations? What are possible alternative interpretations of the data?

Researchers' often arbitrary choices are only one reason (the uncertainty involved in statistics being another; Hodges 1987) for why the "concepts of 'proof' and 'confirmation' are incompatible with science" (Lilienfeld et al. 2015, p. 10). Specifically, evidence does not

lead to sure knowledge; science does not provide absolute proof; scientists are not objective (McComas 1996). Therefore, "no theory in science [...] should be regarded as strictly proven" (Lilienfeld et al. 2015, p. 10). Instead, "the onus of scientists is not to demonstrate that a theory is correct but rather that it is robust under continual scrutiny. Nothing in science is set in stone" (Denholme 2020, p. 121).

However, this is not how science is commonly portrayed in the media. For example, a quick search of the News on the Web corpus (which at the time of writing contained about 17 billion words; https://www.english-corpora.org/now/, accessed on 12 April 2023) shows that the most-common adverbs immediately preceding "proven" are "already", "scientifically" and "clinically", whereas the most-common adverbs immediately preceding "unproven" are "still", "largely" and "yet", falsely suggesting that science can prove claims or theories and possibly suggesting that what is not yet proven could be proven in the future. Perhaps more worryingly, the most-common nouns immediately preceding "prove" and "disprove" in the corpus are "assumptions" and "findings", respectively, where the former suggests a connection between assumptions and proof. Finally, the most-common verbs occurring immediately after the words "study," "research", and "scientists" are "found," "shows", and "say", respectively, again suggesting a level of certainty that is not warranted.

In this context, previous studies (e.g., Frewer et al. 2003; Guenther et al. 2019; Han et al. 2021; Stocking 2010) have shown that, in science reporting, uncertainty is often avoided so as to not "cause distrust in science and scientific institutions" (Retzbach and Maier 2015, p. 432). However, a new study by Wegwarth et al. (2020) suggested that conveying certainty in an inherently new and uncertain situation such as a global health crisis may actually "adversely affect citizens' trust and compliance with containment measures should those reports [on threat scenarios] later prove invalid" (p. 1; cf. Kreps and Kriner 2020). Instead, "well-communicated uncertainty in risk information [...] may be able to achieve important risk communication goals with only limited effects on trust" (Balog-Way and McComas 2020, p. 840), and studies in a Scandinavian context suggest that health experts might even acknowledge uncertainty to boost their credibility (cf. Ihlen et al. 2022; Kjeldsen et al. 2022). In the case of COVID-19, as described above, there was a gradual shift in the messaging, relating to the unknown or yet-uncertain nature of the crisis. Whether this uncertainty is duly reflected in media discourse or whether government officials and authorized experts with their attempts to convey certainty of a monolithic "the science" were uncritically echoed in the news is a central concern of this paper.

### 1.4. The Current Study

In the current qualitative media coverage study, we used a discourse analytic approach (cf. Carvalho 2007; Baxter 2010) to look at the discursive construction of "the science" in the discourse on COVID-19 in two major Austrian newspapers. We employed concepts from Critical Discourse Analysis (Fairclough 2015) that we deemed relevant for the goals of the analysis, with a focus on modality and generic references (see Section 2.4 for more details). We concentrated our analysis on the concepts of herd immunity and (vaccine) efficacy for two reasons. First, both concepts have been subject to uncertainty over the course of the pandemic as the original COVID-19 virus has mutated into numerous variants with differing severities and levels of infectiousness (Lippi and Henry 2021; Ramesh et al. 2021). Second, both concepts have been essential for communicating the efforts of the vaccination programs to the public. We ask the following research questions:

RQ1: How are the terms *herd immunity* and *(vaccine) efficacy* presented?

RQ2: How (un)certain is the language that is being used in connection with the terms *herd immunity* and *(vaccine) efficacy*?

RQ3: What do these linguistic means reveal about how the efforts of the vaccination programs have been communicated to the public?

## 2. Methods

### 2.1. Building the Corpora

To build our corpus for our media coverage analysis, we used the media metasearch engine WISO, which allows accessing and searching articles in German-speaking media. We searched for articles including the keywords "Corona", "COVID19", and "COVID-19" in the two Austrian broadsheet papers *Der Standard* and *Die Presse*. As a timeframe, we chose the period between 1 January 2020 and 31 December 2021, to have an overview of the development of the discourse from the onset of the pandemic to when the vaccination program was already firmly in place. Both papers are considered to be quality newspapers (Kontrast 2018) and are among the most-trusted newspapers in Austria, with 69% and 67% of people in 2021 considering *Der Standard* and *Die Presse*, respectively, to be trustworthy (Gadringer et al. 2022, p. 103). *Der Standard* tends to be read by people on the political left (Gadringer et al. 2022, pp. 64–65), whereas *Die Presse* identifies as conservative and neoliberal (Eurotopics 2019), while both attract readers with higher levels of education (MA 2021). *Der Standard* has a higher readership than *Die Presse*, with approximately 650,000 compared to 350,000 readers (MA 2021) in a country with approximately 9 million inhabitants. We chose these two newspapers because they are considered to be trusted newspapers and cover a rather broad part of the political spectrum. We converted the corpus files into text files so they could be used in a corpus program for concordance analysis.

### 2.2. Selecting Key Concepts

We focused here on the key concepts of herd immunity and vaccine efficacy. For each key concept, we selected a key term. The key terms for vaccine efficacy were *effektiv\** and its synonym *wirksam\** (*effective*), which would find instances of *effective* and *efficacy* (*Effektivität/Wirksamkeit*) and all their lemmata. The key term for herd immunity was *Herdenimmunität* (*herd immunity*).

### 2.3. Extracting Concordance Lists

We extracted concordance lists in WordSmith 7.0 (Scott [1996] 2016) for each key term with 250 characters of text to the left and to the right of each term. We translated quotes selected from the concordance lists for this paper either manually or in DeepL, in which case, we then hand-corrected the translations. Here, we present the English translations of the quotes. We also used Lancsbox (Brezina et al. 2020) to extract the immediate context (five words to the left and five words to the right) in which each key term occurred from the above concordance lists. We manually removed duplicates and instances of key terms occurring in the headlines (which often contained dates or article links or identifiers). We then used the remaining immediate contexts to extract 3-grams (i.e., sequences of 3 words) for each key term. The original German quotes along with translations and lists of the most-frequently occurring 3-grams are available at Supplementary Materials: https://osf.io/a52hk/.

### 2.4. Analyzing Certainty

An important part of our paper consists of a linguistic analysis of certainty. Linguistically speaking, certainty can be relayed in a variety of ways. The most-common way to do so is the use of epistemic, also called expressive, modality, which is to do with "the speaker/writer's evaluation of truth," (Fairclough 2015, p.142), i.e., with the question of how probable they consider something to be. Such modality can, for instance, be expressed either grammatically via modal verbs (such as *may*, *will*, *must*; cf., Simon-Vandenbergen 1997) or lexically through modal adverbs (e.g., *possibly*, *probably*, *likely*; cf., Bailey et al. 2014). According to Goatly (2000), by using low degrees of modality, writers/speakers "are claiming higher status or expertise than the reader, setting themselves up as an 'authority' [...] [T]his expertise and authority/status will be reflected in the degrees of dogmatism or assertiveness with which statements and arguments are made" (p. 90). The scale of epistemic

modality reaches from low probability (*may/perhaps*) to high likelihood (*must/definitely*), with the highest probability being expressed by the complete absence of modality, i.e., by simply stating something as a fact, for example by using the generic present tense and/or a generic referent, as in *Science agrees....*

## 3. Results

### 3.1. Herd Immunity

The Austrian government website oesterreich.gv.at states: "Only if as many people as possible are vaccinated will herd immunity develop" (oesterreich.gv.at 2022). In an interview in May 2020, Rudolf Anschober, then Austria's Federal Minister for Social Affairs, Health, Care and Consumer Protection, specified that "to achieve herd immunity you need 60%, 65% to participate [in getting vaccinated]" (ORF 2020a). A few months later, in November 2020, virologist Herwig Kollaritsch, a member of the Corona Task Force of the Austrian government, stated: "Only with a transmission-blocking vaccine, with which the population is broadly vaccinated, can herd immunity be achieved" (Volksblatt 2020). Achieving herd immunity was clearly framed in the official narrative in terms of getting as many people as possible to be vaccinated rather than through immunity through previous infections. In March 2020, Kollaritsch proposed: "We cannot count on natural herd immunity because it is unclear how long after infection people will be immune to the coronavirus. In addition, it could mutate and make the same people sick again. [...] So, vaccination is currently the only way out to fight the pandemic. [...] There are no alternatives in the current situation." (Kurier 2020b).

Since achieving herd immunity by means of vaccination became one of the major goalposts of the pandemic response and a core part of the official narrative, the term is frequently mentioned in the media. The term *Herdenimmunität* (herd immunity) is mentioned 302 times in the *Presse* corpus and 146 times in the *Standard* corpus. A look at the most-frequently occurring 3-grams in the immediate context of *Herdenimmunität* from the *Presse* and *Standard* corpora suggests that herd immunity is presented as a goal (*Ziel*) that needs to be reached (*erreichen, erreicht*) in both corpora.

In the reporting on how to achieve herd immunity, both *Der Standard* and *Die Presse* often do not convey the uncertainty around herd immunity and how to achieve it. This uncertainty stems from the facts that COVID-19 turned out to be a fast-changing virus (Lippi and Henry 2021) and that the vaccines were found to not stop transmission of the virus (Stokel-Walker 2022). Statements implying certainty come both from reporters writing for *Der Standard* or *Die Presse*, as well as from quotes of experts that *Der Standard* and *Die Presse* consulted or quoted. A common example in both the *Presse* (Examples 1–7) and the *Standard* (Examples 8–9) corpora are statements referring to a certain immunity or vaccination rate that *must* be reached for herd immunity, implying a level of uncertainty in our knowledge that is not warranted:

1. "To eradicate the virus, society **must** achieve the much-cited herd immunity. For this virus, the benchmark is 60 percent of the population" (*Die Presse*, 20 March 2020).
2. "60 percent of the population **must** be infected to reach this state [herd immunity]" (*Die Presse*, 26 March 2020).
3. "Thus, 50 to 60 percent **must** be immune for so-called herd immunity to be achieved and for the coronavirus to stop spreading. This is also the value assumed by the World Health Organization (WHO)" (*Die Presse*, 26 November 2020).
4. "One way or another, we will reach herd immunity, for which around 85 to 90 percent of the population **must** be immune, in the fall. Those who do not get vaccinated will most likely be infected and get their immunity this way" (*Die Presse*, 18 April 2021).
5. "For this [herd immunity] to happen, at least two-thirds of the population **must** be immunized" (*Die Presse*, 1 June 2021).
6. "To prevent strong waves of infection, 85 percent of the population **must** be immunized" (*Die Presse*, 24 July 2021).

7.  "For herd immunity to be achieved, about 85 to 90 percent of the population **must** be vaccinated or recovered" (*Die Presse*, 11 August 2021).

8.  "This would be sufficient for so-called herd immunity, in which at least 60 percent of the population **must** be vaccinated" (*Der Standard*, 21 December 2020).

9.  "For a pandemic to end, herd immunity to the pathogen is needed. To achieve this, experts estimate that 70 to 80 percent of a population **must** be immune. The vaccines currently available have an efficacy of 62 to 95 percent, depending on the study and the active ingredient" (*Der Standard*, 13 March 2021).

Despite the certainty conveyed in the above examples, there is no agreement as to how many people in a population *must* be vaccinated to achieve herd immunity. The stated percentages range from 60 percent to 90 percent of the population. Readers are thus confronted with inconsistent information that is presented with a level of certainty that is questionable. Other frequent terms conveying certainty are *is/are needed* and *is/are necessary* or *only*. The *Presse* (Examples 10–13) and *Standard* (Examples 14–15) corpora contain the following examples:

10. "The end of the crisis is through vaccination—and through vaccination skeptics. Six million vaccinated persons [about 66%] **are needed** in Austria for the phenomenon of herd immunity" (*Die Presse*, 14 December 2020).

11. "Even the most comprehensive and objective information campaign on the COVID vaccination will not be sufficient to motivate **the number** of citizens **necessary** for herd immunity to get vaccinated. The current level of personal suffering is too low and the uncertainty caused by the counter-campaign of the opponents of vaccination will be too strong" (*Die Presse*, 31 December 2020).

12. "In addition, a comprehensive vaccination campaign **is needed** to quickly get close to herd immunity. The **only** way out of this pandemic is vaccination" (*Die Presse*, 25 March 2021).

13. "For the unvaccinated and unimmunized, restrictions **are necessary** until enough people have been immunized and the phenomenon of community protection occurs, also known as herd immunity" (*Die Presse*, 8 September 2021).

14. "Now there is the problem that to achieve herd immunity, a vaccination coverage rate of 60 to 70 percent **is needed**" (*Der Standard*, 9 December 2020).

15. "However, at least 70 percent **are needed** to establish herd immunity in a country" (*Der Standard*, 11 December 2020).

Wording implying certainty is even found in the context of future predictions, where certainty cannot be expected. Examples come from of the use of *will* rather than the less certain *would*, as in the following from the *Presse* (examples 16–17) and *Standard* (Example 18) corpora:

16. "Is Sweden playing Russian roulette with coronavirus? Herd immunity. According to forecasts, more than half of the Swedish population **will be** infected with COVID-19 by the end of April" (*Die Presse*, 3 April 2020).

17. "Because in vaccination economics, there is only black and white: economies that finish the race for herd immunity first would be rewarded with strong economic multiplier effects as early as the second half of the year, while the EU will likely be grounded in crisis mode until 2022 at the prevailing pace of vaccination and **will be** confronted with significant costs" (*Die Presse*, 10 February 2021).

18. "Vaccination is currently the most powerful tool that policymakers have in their hands to contain the pandemic. However, herd immunity against the coronavirus **will only be** achieved if 80 to 85 percent of the people are vaccinated" (*Der Standard*, 14 August 2020).

Wording implying a lower degree of certainty is less commonly found in the corpora in connection with herd immunity. One example comes from the *Standard* corpus, where *would be* implies a certain amount of uncertainty: "The best vaccine is ineffective if it is not vaccinated. Only about half of the people in Austria want to be vaccinated against COVID-

19, 60 to 70 percent **would be** necessary for herd immunity" (*Der Standard*, 5 December 2020).

Counts of words conveying certainty vs. uncertainty about present and future events in both the *Presse* and *Standard* corpora support this impression, when looking at the immediate context of *herd immunity*. In the *Presse* corpus, the words *ist* and *sind* (is/are) occur 32 and 5 times, respectively, in the immediate context of herd immunity, whereas the words *wäre* and *wären* (would) occur only 12 and 2 times in this context. Similarly, the words *wird* and *werden* (will) occur 27 and 19 times, respectively, but the words *würde* and *würden* (would) only 3 and 1 times in the immediate context of herd immunity. In the *Standard* corpus, the words *ist* and *sind* (is/are) occur 16 and 11 times, respectively, in the immediate context of herd immunity, but the words *wäre* and *wären* (would) each occur only once. The words *wird* and *werden* (will) occur 16 and 11 times, respectively, but *würde* and *würden* (would) occur only 4 and 0 times in the immediate context of *herd immunity*.

### 3.2. (Vaccine) Efficacy

Vaccine efficacy was employed as one of the main arguments within the official narrative for the mass vaccination program, as well as for reaching the goalpost of herd immunity. For instance, in a 27 December 2020 article on the national broadcasting corporation ORF's website, the head of the National Vaccination Board, Ursula Wiedermann-Schmidt, was reported as saying, "The efficacy of vaccines directly affects the extent of the vaccination rate necessary to achieve herd immunity" (ORF 2020b). The initial claims made by the pharmaceutical companies responsible for the first vaccines ranged between 95% for Biontech/Pfizer's mRNA-vaccine and 66% for Johnson and Johnson's vector vaccine (APA 2021). Even though with time, these numbers were amended downwards, efficacy continued to be one of the buzzwords in the government's campaign to get more people vaccinated, which was postulated to be the only way to end the pandemic. In August of 2021, for instance, Austrian Chancellor Sebastian Kurz called on Austrians: "The COVID-19 vaccine is our only way out of the pandemic, and it protects from infection—and by extension also from long-haul COVID—by more than 90%" (Bundeskanzleramt 2021a). Even when reports started to emerge of waning vaccine efficacy, government and health officials insisted that mass vaccinations were the way to get out of the Corona pandemic. In September 2021, Chancellor Kurz was reported as saying, "Vaccine protection wanes with time, which is why it is important to get the third shot in due time after the second [...] 'The third shot should, however, not be at the discretion of each individual: Only the third shot provides long-term protection. Otherwise, you will be at the mercy of the virus, which is particularly dangerous for the elderly population.'" (Bundeskanzleramt 2021b).

An analysis of the corpus showed a large number of entries for *effective/efficacy* and their respective lemmata (in German: *effektiv/Effektivität/wirksam/Wirksamkeit*), namely more than 1700 times—697 times in *Der Standard* and 1011 times in *Die Presse*.

A look at the most-frequent 3-grams suggests that the words *efficient* and *efficacy* frequently co-occur with the terms *safe* or *safety*, in both corpora. This phrase was continuously repeated in media discourse as a given fact, corroborated with empirical data. Let us consider how often the term *efficacy* is quantified with percentages that suggest absolutes, e.g., that vaccines have more than 90% efficacy against the Coronavirus. To a layperson, this number is likely to suggest an absolute scale, i.e., that the vaccine is more than 90% effective in preventing infection, symptoms, and transmission. However, that is not what these numbers meant. Rather, they denoted the *relative* likelihood of contracting the virus compared to an unvaccinated person. In the original Pfizer trial, for instance, out of approximately 20,000 vaccinated people, 9 people contracted COVID-19, while 162 out of roughly the same number of unvaccinated people in the placebo group became infected. The difference between 169 vs. 9 infected was then calculated to mean 95% efficacy (Polack et al. 2020). In absolute numbers, the vaccine would have been 99.95% effective; however, by the same token, being unvaccinated could have been interpreted as being more than 99% effective against infection. According to Malhotra (2022), this means

that taking a vaccine would merely provide 0.84% in *absolute* risk reduction, far off the popular claim. By focusing on the relative numbers, the media arguably can be thought of having misinformed their readers into believing that taking a vaccine had a much greater advantage than the absolute numbers warrant.

The most-frequently occurring 3-grams also suggest that the depiction of efficacy tended to be somewhat inconsistent in media reporting: some reports refer to efficacy "against the virus"; others mention "efficacy against infection", while yet other limit efficacy to "the vulnerable groups." This becomes an issue of interest if we consider that words like *efficacy* are likely to raise expectations in news readers, and when those are not fulfilled, this might lead to media consumers being more critical of media and their coverage of scientific issues. In line with this suggestion, according to a survey in Austria, public trust in the media has decreased by 11% from mid-2021 to mid-2022 (OGM 2022), and a study from Germany suggests that 41% of people think that the credibility of journalism has declined as a result of Corona reporting (TU Dortmund 2022).

Except for critics of the vaccines, which were featured sparsely in the corpus, the terms *efficient* of *efficacy* were mostly used in the affirmative, i.e., they were used to state, maintain, and stress the efficacy of the COVID vaccines. Only later in the process, as waning efficacy became a point of concern, did the term start to appear in negative contexts, but always with an affirmative reference regarding the increased efficacy of the so-called booster shots. The following are examples of such mentions, some of which are explicit, while others are implicit through presuppositions:

19. "**The extraordinary efficacy** of the vaccines increases the benchmark and decreases the circle of candidates who can be successful." (*Die Presse*, 12 Dec ember 2020).
20. "This vaccine, with its efficacy of more than 90 percent, **belongs to the most effective vaccines** of all times." (*Der Standard*, 6 September 2021).
21. "Both available mRNA-vaccines **are not only highly efficient**; the speed by which they reached marketability has even stunned experts." (*Der Standard*, 24 August 2021).

The definite wording of the extracts above, which are representative of much of the instances of use of the search term, shows a high degree of certainty at a time when really nothing was all that certain yet, as all vaccines had been researched, tested, and produced in the fast lane. There is also a tendency towards constructing the claims by pharmaceutical companies, as well as government officials and their official medical experts as a general consensus, as "the science", thereby giving news consumers a distorted picture of scientific discourse:

22. "The authorized vaccines by Biontech/Pfizer and Moderna are amongst the most efficient and well-tolerated vaccines ever to be developed. **Virologists and infectiologists** are in complete agreement about that." (*Die Presse*, 22 January 2021).
23. "**Science** agrees that the most efficient means by far would be the vaccine." (*Die Presse*, 11 January 2021).
24. "**Science** has quickly developed a vaccine against the disease, whose efficacy is **undisputed** and publicly visible." (*Der Standard*, 17 September 2021).
25. "The intermediate results of the Phase-III study show that the vaccine is more than 90% effective, which is **unanimously** seen as a great success by **independent experts.**" (*Der Standard*, 11 November 2020).

Again, the use of certainty implies factual information, whereas the generic references suggest unanimous agreement by the entire medical field. However, from the onset of the pandemic, there has been disagreement amongst medical doctors and experts, many of which went public with opinions that differed from and sometimes contradicted the dominant narrative (e.g., Kory et al. 2021; Kulldorff et al. 2020; Malhotra 2022; McCullough et al. 2020). Their opinions remain unrepresented in *Der Standard* and *Die Presse*; the reference to what seems like a monolithic narrative, however, could be interpreted as an implicit, but sweeping delegitimization of non-conformist approaches. The discourse also

seems to ignore that laboratory data can be biased and appears to evaluate the existence of such data as irrefutable proof for the excellent efficacy of the vaccines.

The certainty extended also to the discussion of vaccine efficacy on possible variants of the virus:

26. "Could the mutation reduce the efficacy of the vaccines? **Most virologists** do not share this assumption. For a simple reason: The available vaccines and those being close to authorization **do not just detect one, but various (in fact all) parts** of the spike-protein on the surface of the virus." (*Die Presse*, 22 December 2020).

27. "The fear that the new variant could strongly reduce the efficacy of the available vaccines **is unfounded**, say **most health experts**." (*Die Presse*, 8 January 2021).

28. "According to a study by the US pharma company Pfizer, the vaccine produced in cooperation with the German company Biontech also **protects** against the coronavirus mutations that are dominant in Great Britain and South Africa. Antibodies from the blood of 20 vaccinated individuals **were** 95% **effective** against 16 mutations. Whether further mutations in the viral DNA can make the authorized vaccines ineffective is unclear, but **improbable** according to the virologist Christian Drosten from Berlin." (*Der Standard*, 12 January 2021).

When news of waning efficacy and failure to prevent transmission and infection eventually did start emerging, there was a gradual change in the narrative. Originally, the government, as well as the news media had insinuated that the vaccine provides protection from infection as well: "Those who are vaccinated will not get infected with the virus on renewed contact—at least with a probability of more than 90%, corresponding to the efficacy indicated for the vaccines." (*Die Presse*, 22 January 2021). *Der Standard* had maintained the same claim: "People who have received two shots of the vaccine had a 92% lower risk than unvaccinated people to even get infected with the virus according to the study. This also reduces transmission." (*Der Standard*, 26 February 2021). Increasingly, however, the focus became that (1) the vaccines are still effective, that (2) no vaccine is perfect, and that (3) the original purpose of the vaccine had been to protect against severe disease and death only:

29. "During the authorization procedure, a vaccine is tested on its risk-benefit ratio. **Neither 100% safety nor absolute efficacy are required, and nor are the manufacturers claiming this**. Ultimately, it is the manufacturer who is liable irrespective of authorization. (*Die Presse*, 13 March 2021).

30. "A lab study by Biontech/Pfizer concludes that the South African variant reduces the neutralization rate of the vaccine by about two thirds. **Nevertheless, scientists believe in its efficacy.**" (*Die Presse*, 19 February 2021).

31. "The vaccines are also **effective** against mutations. Vaccination expert Herwig Kollaritsch **reaffirms the efficacy and safety** of the Coronavirus vaccines." (*Die Presse*, 21 January 2021).

32. "Tobias Welte, director of the clinic for pneumology at the Medical University of Hannover and former president of the European Respiratory Society **emphasizes that the vaccines are also effective against the growing Delta variant** in Europe." (*Die Presse*, 6 October 2021).

33. "The **main job of a vaccine consists of protecting the body after infection** from falling ill and in particular from **a serious case of the disease**." (*Die Presse*, 12 May 2021).

It is noteworthy that the significant drop in the efficacy of COVID-19 vaccines does not seem to have led to a re-evaluation of the original estimates or a reflection on the potential overselling of the power of the vaccines in question; on the contrary, the data we analyzed showed a reaffirmation of their importance and to the propagation of a third shot. Initially, though, government, pharma companies, and medical experts had insisted that 2 shots were sufficient to protect vaccinees (1 shot for Johnson&Johnson), e.g.,: "A second shot is necessary to obtain full efficacy." (*Die Presse*, 29 April 2021). Now, seemingly without

revisiting the previous claim with some scrutiny, the narrative shifted towards suggesting that a three-shot-regimen was not only necessary, but also the norm:

34. "**Many other well-established vaccines**, too, have to be administered three times to be sufficiently efficacious." (*Die Presse*, 11 November 2021).

35. "A similar reduction in efficacy can be detected in the Pfizer vaccine. **A third shot** of both vaccines **returns efficacy** to a level of 70 to 75%." (*Die Presse*, 14 December 2021).

36. "According to the study, which was published in the New England Journal of Medicine, efficacy sank to 70%. In the case of Delta, it was 93%. Health experts **urgently recommend a booster shot**, which significantly increases protection." (*Der Standard*, 31 December 2021).

What is interesting in this respect is the use of the term "*mmunized* (German *immunisiert*). Its use seemed to imply complete protection from the virus, but even when breakthrough infections increasingly became the norm, our corpus shows that the term continued to be used, even for individuals who had only received two shots:

37. "**Even fully immunized people have been infected**. How is that possible? Answer: This is not at all surprising, as **no vaccine is 100% efficient**." (*Der Standard*, 30 July 2021).

38. "[...] that new numbers indicate that the **protection from infection has sunk to 40%**. The number **denotes fully immunized individuals**, i.e., those that have already received the two necessary vaccine shots." (*Der Standard*, 26 July 2021).

39. "'The **infected person** was **fully immunized**,' said a press release by the festival. This means she had received both vaccine shots and had nonetheless become infected—a rather rare occurrence." (*Der Standard*, 20 July 2021).

40. "'We know that it **is necessary to get a booster**, a third shot,' said Ludwig. The booster, he said, serves a more efficient protection against the virus, in particular against the Delta variant. In Vienna, 64% of people currently have received the first shot, 61% **have full protection**." (*Die Presse*, 15 October 2021).

Clearly, the use of *immunized* in this context can be very misleading. Up until the COVID-19 pandemic, the definition of *immunized* used to be "to give (someone) a vaccine to prevent infection by a disease" (Merriam-Webster 2016). Only in 2021, the definition changed to "to make (a living organism) immune or resistant to a disease or pathogenic agent especially by inoculation" (Merriam-Webster 2021a), thus allowing for a looser definition of the term. This goes hand-in-hand with a change of the definition of vaccine from meaning "a preparation of killed microorganisms, living attenuated organisms, or living fully virulent organisms that is administered to produce or artificially increase immunity to a particular disease" (Merriam-Webster 2021b) to denoting "a preparation that is administered (as by injection) to stimulate the body's immune response against a specific infectious agent or disease" (Merriam-Webster 2021c) within a week's span in January 2021. It is not clear, however, if average news consumers were aware of this change in definition. They would, therefore, be more likely to interpret it as making someone immune from infection and to have thought of vaccines in much the same manner.

One final point regarding the use of efficacy is linked to the original purpose of the vaccines, which was to protect the vulnerable segments of the population from the virus, as in this example from the *Standard* corpus: "This vaccine has achieved an efficacy of 94% in the group of 65-85-year-olds. It is therefore also efficient in those people who most urgently need to be immunized." (*Der Standard*, 27 November 2020). When the rate of hospitalization fell, this was seen to be a result of the excellent efficacy of the vaccine for these vulnerable groups:

41. "The efficacy of the vaccine is illustrated by looking at patients in intensive care. During the entire course of the pandemic, their average age was 66 years, but in July 2021 it sank to 61 years—**a result of the higher vaccination rate** in elderly people." (*Der Standard*, 9 September 2021).

When, however, such groups were increasingly affected by infection, hospitalization, and death, this was suggested to be a consequence of their immunocompromised condition. This interpretation was absent from the unvaccinated group, whose status as ICU patients was implicitly attributed to their being unvaccinated rather than to any existing pre-conditions or comorbidities:

42. "A survey conducted in Israel has recently shown that the **very risk groups that had a high risk of serious disease** without a vaccine, **also had an increased risk for a serious breakthrough infection**: the elderly and people with comorbidities." (*Der Standard*, 31 July 2021).

43. "[In the province of Salzburg] 24 beds in ICUs are occupied by COVID patients. **Eight of those are vaccinated, all of which are immunocompromised** due to chemotherapy or organ transplants." (*Die Presse*, 13 November 2021).

44. "Of the 16 ICU-patients, one was fully immunized, twelve were unvaccinated, one was partly immunized, and **two were fully vaccinated, but immunocompromised**." (*Die Presse*, 21 August 2021).

Therefore, the line of argument was reversed from "vulnerable groups need to be vaccinated because their immune system is compromised" to "vaccines don't work well for them because they are immunocompromised". This and other apparent inconsistencies in the narrative—and the failure of governments and media to meaningfully explain them—can arguably have the potential to create great confusion and distrust in science and scientific processes and can be suggested to be one of the driving forces for the rise in vaccine hesitancy.

## 4. Discussion and Conclusions

In this study, we analyzed the discursive construction of "the science" in the discourse on COVID-19 in two Austrian newspapers, with a focus on the concepts of herd immunity and vaccine efficacy. In this context, we observed that the definitions for all concepts involved were subject to change over time. While the definition of "immunized" shifted between August 2016 and February 2021 from "being immune" to including "being protected" from a disease, the concept of vaccination equally changed. On January 18 2021, it was still defined as a preparation "administered to produce or artificially increase immunity"; one week later, on January 25, the definition was modified to "a preparation that is administered (as by injection) to stimulate the body's immune response" against a disease (cf. Merriam-Webster 2016, 2021a, 2021b, 2021c). Similarly, the goalposts for the pandemic response kept shifting: whereas in 2020, the benchmark to reach herd immunity was set at 50–60%, by August 2021, the number had risen to 85–90%. In the same period, the purported efficacy of the COVID-19 vaccines as discussed in our corpus moved from 95% at the beginning of 2021 to 70% and below by the end of 2021. Despite the fluid and mercurial nature of all these benchmarks, goalposts, and concepts, overall, we found that the newspapers analyzed overstated how certain we are in "the science" and in our current knowledge about herd immunity and vaccine efficacy. Our results were, thus, in line with Guenther et al.'s (2019) findings, in the context of German print and online media, that "scientific findings are predominantly depicted as scientifically certain".

The discourse on herd immunity focused on herd immunity as a goal that needs to be achieved through immunity of a certain percentage of the population (RQ1). Herd immunity is an abstract concept and is in itself not directly measurable. To use herd immunity as guidance during the pandemic means to rely on estimates rather than on actual numbers that we can measure with more or less accuracy, such as the number of infections, severe cases, deaths, recoveries, etc. Herd immunity is also a moving target, as the differing estimates for achieving herd immunity over the course of the pandemic illustrate. Despite this, Austria and many other countries in the Global North have adopted the concept of herd immunity as an overarching goal that needs to be achieved and have made very concrete claims about how many people need to be immune to achieve herd immunity.

Efficacy, likewise, is rather abstract as a concept, but very successful as a buzzword. Bare statements like "safe and effective" imply absolute effects, when in fact they are being used to illustrate relative phenomena (RQ1). When the media and officials state that the vaccine is more than 90% efficient, they do not mean to say that the risk of contracting COVID is less than 10%; they mean to say that being vaccinated makes you 90% less likely to get COVID compared to being unvaccinated. This sounds like a significant effect. However, when looking at the absolute numbers in the original Pfizer trial (cf., Polack et al. 2020), it quickly becomes clear that being unvaccinated also accords individuals with a 99%+ efficacy of not contracting COVID. By presenting these percentages out of context, it can be argued that the media have misinformed their audiences into believing the vaccines are much more powerful than they eventually turned out to be. In particular, seeing that the original goals of the vaccination program were to stop transmission and stop the disease (WHO 2020c, 2021), the fact that the vaccines ultimately turned out not to be designed to do that—in fact, they were never tested to do that, as a Pfizer executive had to admit at a hearing in the EU parliament in late 2022 (Chung 2022)—was not explicitly reflected on in our corpus.

What the corpus data do show is a consistent use of certainty in reporting (RQ2, cf. Guenther et al. 2019) on what eventually turned out to be a constantly changing pool of knowledge on the nature of the Coronavirus and how to confront it. We argue that this certainty was the result of taking sources (official government sources and officially approved experts) at face value and, thus, naturalizing their narratives, rather than being skeptical or inquisitive on the matter by, for instance, introducing and giving space to expert voices outside the sanctioned spectrum. The discourse very much suggested that the current knowledge and measures propagated by officials and their medical experts was not only state-of-the-art, but also the only possible conclusion from the data available. Science was therefore rendered in absolute terms—becoming "the science"—rather than a constant work in progress where thesis meets antithesis to become synthesis (RQ3). It was the impression that we got from our corpus that there was only ever a thesis, and any antithesis that may have existed was not actively part of the dialectic discussion of the topic in the media outlets we analyzed. This led to some unforeseen and possibly unfortuitous outcomes in that, as the narrative kept shifting, the discrepancy between what media and officials had postulated at the outset of the pandemic and then at later times became increasingly apparent. It can be argued that, if more scientific theories or antitheses had been introduced into the discussion (cf. Balog-Way and McComas 2020; Ihlen et al. 2022), no such discrepancy would have been observable, or at least not to the present extent and not as antagonisms and contradictions. By including different theories and voices, the work-in-progress nature of science would have been more apparent; the changing narrative would have been perceived as the temporary synthesis of an ongoing process. However, due to the dogmatic nature of the media discourse that we observed in our data, the sudden changes in the narrative were much more sudden and salient.

The media's tendency to present information as "the science" might have backfired as the public have experienced the work-in-progress nature of science first-hand during the pandemic and may have become wary and/or suspicious of the certainty with which the media portrayed information that turned out to be uncertain and ever-changing. One of the problems is that governments and the media, and to some extent scientists themselves, have promoted the idea that we have certain knowledge about various aspects of the pandemic, possibly in an attempt to reassure the public (cf., Retzbach and Maier 2015). Instead, our knowledge during the pandemic was ever-evolving and -changing, and the—sometimes dogmatic—certainty with which these two Austrian media outlets expressed the extent of our knowledge and capabilities may have led a substantial portion of the Austrian population to lose trust in the COVID-19 media coverage. Overall, the corpus data suggest that, indeed, the media have closely followed, observed, and echoed the official narrative of the pandemic, rather than critically questioning or scrutinizing it when this was warranted. The consistent use and high degree of certainty that the two selected news outlets employed

for something that was and is in constant flux was to us the clearest indicator of this new role of the media—as mediators *for* power, rather than as a counterweight *to* power.

**Supplementary Materials:** The original German quotes along with translations and lists of the most-frequently occurring 3-grams are available at: https://osf.io/a52hk/.

**Author Contributions:** Conceptualization, J.S. and A.F.; methodology, J.S. and A.F.; formal analysis, J.S. and A.F.; investigation, J.S. and A.F.; data curation, J.S. and A.F.; writing—original draft preparation, J.S. and A.F.; writing—review and editing, J.S. and A.F.; visualization, J.S. and A.F.; project administration, J.S. All authors have read and agreed to the published version of the manuscript.

**Funding:** This research received no external funding.

**Institutional Review Board Statement:** Not applicable.

**Informed Consent Statement:** Not applicable.

**Data Availability Statement:** The data presented in this study are available on request from the corresponding author. The data are not publicly available due to paywall restrictions.

**Acknowledgments:** The authors acknowledge the financial support by the University of Graz. We would also like to thank Katharina Haslacher for her help with building, cleaning and searching the corpus and formatting the paper.

**Conflicts of Interest:** The authors declare no conflict of interest.

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
