# Peer review of "The Use of Certainty in COVID-19 Reporting in Two Austrian Newspapers"

_journalmedia, doi:10.3390/journalmedia4020033_

Round 1

Reviewer 1 Report

Thank you for the article. I enjoyed reading it, but I still think there is a need for improvements. Please find my comments below. 

Introduction, on p. 1 you claim that you explore how the Austrian media deal with information about aspects of Covid-19. I think you should correct this to how two Austrian media deal with information…. You cannot make generalisations about patterns based on a few numbers.

Media and media sources, on p. 2 I would like to have a reference (source) to the claim that communicating the goals of WHO and the ministry of health and how to achieve them became the tasks of the news media. Is it not mainly a government communication task? (see section II in the book Communicating a Pandemic, particularly chapter 10 about the dual roles of the news media as watchdogs and government megaphones).  I also think you should drop the reference to the “complex definition of newsworthiness” as this is not a topic in the analysis/ discussion.

Expert sources, on p. 3: I would like to recommend research done by professor Ihlen and professor Kjeldsen about expert ethos during the pandemic. I think these references are more appropriate than the paragraph about the study on the Syrian Conflict. You could also add a literature review from framing studies of the pandemic, enhancing the use of sources (Cho and Wang 2021; Hart et al. 2020; Tejedor et al. 2020; Fonn and Hyde-Clarke 2021; Hubner 2021).

Literature review: Perhaps you could refer to findings from similar previous text/discourse analyses based on similar methods (see some suggestions at the end of this document).

Methods, p. 5 It will help the readers if you provide us with some reflection regarding the choice of these two newspapers and the period. What kind of newspapers are these two? Are they representatives of the news media in Austria? Also, how do you know that “the discussions about the key concepts changed over the course of the pandemic”?

More importantly, I believe the study will benefit from a more in-depth presentation and discussion of methods than the ½ page does. To me, as a media scholar, it does not come across as a “media coverage study or a media coverage analysis. Is it not a discourse analysis? Or a computational content analysis? Text mining study? A linguistic analysis? The study “Collocation analysis of news discourse and its ideological implications” by Huei-ling Lai might be an inspiration.

Could you please explain what you mean by a “discourse analytic approach”? Is it discourse analysis as in Chomsky’s view? Fairclough?

The analysis: Overall, the description of your findings is very detailed and rich, could you provide an overarching paragraph about how the narrative developed and shifted, including how and when definitions have changed?  

on p. 11, p. 12 and p. 18 please reconsider speculating about whether the media has “misled their readers” and whether “media consumers might become more critical of coverage on scientific issues, in addition to “smaller loss of credibility in the media”. Which newspaper do you refer to for these incidents? How often? And in what context? Personally, I don’t think such speculation should be done in the analysis. You could perhaps discuss them in the discussion section.

p. 13 Source for the claim that “there has been widespread disagreement amongst medical doctors and 558 experts, many of which went public with opinions that differed from, sometimes contradicted the dominant narrative”?

Discussion: The study’s implication and theoretical contribution would improve if you discussed your findings considering other, similar studies. There is a newly published book from Nordicom that would be useful for this purpose.

I’m not sure you have enough data supporting the claim that the media had a “strategy of presenting information as “the science” (p. 19). How do you know what the intention of media executives and journalists were? Was the strategy the same in both newspapers?

References:

 https://www.nordicom.gu.se/en/publications/communicating-pandemic?fbclid=IwAR0hK50wnhuhNdMgU52tu0cwY0Qr4vCTgfzPoSJ8k26MMQ5LUqttfiu2E1U

Kjeldsen, J. E., Ihlen, Ø., Just, S. N., & Larsson, A. A. O. (2022). Expert ethos and the strength of networks: negotiations of credibility in mediated debate on COVID-19. Health Promotion International37(2), daab095.

Ihlen, Ø., Just, S. N., Kjeldsen, J. E., Mølster, R., Offerdal, T. S., Rasmussen, J., & Skogerbø, E. (2022). Transparency beyond information disclosure: strategies of the Scandinavian public health authorities during the COVID-19 pandemic. Journal of Risk Research, 1-14.

Lai, H. L. (2019). Collocation analysis of news discourse and its ideological implications. Pragmatics29(4), 545-570. https://www.jbe-platform.com/content/journals/10.1075/prag.17028.lai

Nor, N. F. M., & Zulcafli, A. S. (2020). Corpus driven analysis of news reports about Covid-19 in a Malaysian online newspaper. GEMA Online Journal of Language Studies20(3), 199-220.

Ng, R., & Tan, Y. W. (2021). Diversity of COVID-19 news media coverage across 17 countries: The influence of cultural values, government stringency and pandemic severity. International Journal of Environmental Research and Public Health18(22), 11768.

Reviewer 2 Report

This is a very interesting, relevant, and original paper. It is an important interdisciplinary contribution to journalism studies and discourse studies. 

However, I have some comments. First, the author(s) write that they use a discourse analytic approach to study the discursive construction of the concept science. I think the theoretical background lacks at least a brief review of other papers published about the linguistic expression of (un)certainty in journalistic writing, media texts, etc. that have used discourse analytic approach and relevant discourse / linguistic theories in their research on (un)certainty. This could be included in-between sections 1.3 and 1.4 or in section 1.4. 

The second comment concerns the use of English articles. I think in line 34, the indefinite article a is not necessary ("[...] distributed by a media that had come to see themselves [...]"). Besides, the definite article the seems unnecessary when the authors mention the term science as in lines 221 ("[...] their attempts to convey certainty of a monolithic "the science" were uncritically echoed in the news."). See also line 14 (in the examples provided below, science is not used with the definite article). In 4. Discussion and Conclusion, the science is used throughout. The author(s) should be consistent in their use of the definite article with the concept science

Thirdly, please check grammar in line 553 ("[...] the terms efficacy is quantified [...]"). 

Next, it is preferable to be consistent in the use of terms. In line 260, the author(s) introduce the term "vaccine effectiveness", though throughout the paper they use the term "vaccine efficacy". 

Lastly, the subjective evaluation by the authors (line 825) of the work of editorial boards of dictionaries is questionable. As a rule, any new entries and modifications of definitions are discussed in detail by lexicographers, terminologists, linguists and other specialists as part of their daily work. True, the discussions are not public, and subjectivity plays a role in human decision-making, but it does not seem correct to claim, without providing any evidence, that something was changed "quietly", in some unusual, as if conspiratorial, way in this particular case. Similarly, the author(s) refer to "the lack of humility in the Austrian media". Again, this sounds too subjective, maybe the word humility is not the right one to express what the author(s) indeed meant. 

Round 2

Reviewer 1 Report

Thank you for the reviewed article. Overall, I think it has improved after the first round of reviews and the below suggestions for minor changes are associated with:

-          1) Interpretation and conclusions – your claim about the new role of the media as mediators for power - is warranted by and sufficiently derived from/focused on the data.

-          2) Key messages/What this paper adds – reflect accurately what the article says and what topics you can’t say something about (for instance trust). Practical implications?

-          3) discourse analysis – more specific about the analytical design and the discursive construction/ narratives

Abstract

I really value how you point out key findings (the overuse of items implying certainty and the changing roles of the watchdog to the mediator).  However, I am not sure you are entitled to claim that this decrease “the public’s trust in the accuracy of the Covid-19 information presented in the media”. This is mere speculation by the authors.

Part 1 Introduction

This section has a better focus now, but the argument based on Wegwarth et al., 2022 that policymakers and health experts sometimes shy away from communicating scientific uncertainty needs more support and/or discussion. Research from my own country suggests the opposite,

-          Kjeldsen, J. E., Mølster, R., & Ihlen, Ø. (2022). Expert uncertainty: Arguments bolstering the ethos of expertise in situations of uncertainty. In The pandemic of argumentation (pp. 85-103). Cham: Springer International Publishing.

-          Ihlen, Ø., Just, S. N., Kjeldsen, J. E., Mølster, R., Offerdal, T. S., Rasmussen, J., & Skogerbø, E. (2022). Transparency beyond information disclosure: strategies of the Scandinavian public health authorities during the COVID-19 pandemic. Journal of Risk Research25(10), 1176-1189.

In these studies, the authors argue that health experts acknowledge uncertainty to bolster their credibility, and as such, may not have negative effects on people’s trust in the communicators. The studies also provide you with studies backing your argument that such communication can be risky.

Moreover, I don’t understand the argument that “it stands to reason that newspapers no longer see themselves as a corrective will uncritically echo….” P. 2 (there is also a “that” too much). I think this is a harsh comment about the professionalism of journalism. To my knowledge, journalists worldwide take the challenges associated with fake news, fact-checking, propaganda etc seriously. Perhaps you could add some studies supporting your argument of the “selective echoing press”.

Part 1.1 Media, sources, and the use of certainty

This section has also improved. I think it was the right decision to delete chunks of text not directly relevant to the analysis. Perhaps you could give more space to the notion of the accountability role of the news media, and whether this is a more important function during the pandemic than under normal situations.  

Part 2 Methods

Please elaborate on what kind of “discourse analytic approach” you are using and how this relates to the very uncertain context of the pandemic.

Part 2.4 Analyzing certainty

This is a good paragraph. It is important to define this very relevant term. It would be useful to highlight the difference between expert alternative voices outside the authoritative governmental sphere, as you later claim that the media does not give space to such voices outside the “sanctioned spectrum”.

Part 3 results

Reducing the descriptive text has made this section more interesting and to the point. I’m not convinced about the discourse analytic approach. Which discourses have you identified? What are their functions? Please elaborate on and show how exactly “the corpus data suggests that indeed the media have closely followed, observed and echoed the official narrative of the pandemic, rather than critically questioning or scrutinizing it when this was warranted”. Also, you need to be specific about the findings of “the changing roles of the watchdog to the mediator” – how and when did this change occur?

Part 4 Discussion and conclusion

I would very much see a closer discussion of the research question in relation to the literature review on both media’s role, sources, and the use of scientific information. Sometimes I get a feeling that you are shooting the messenger and accusing the media of misinforming the audience purposely. The text comes across as speculative. Perhaps it is more fruitful to discuss the media’s use of sources and not the vaccination program per se. I relation to RQ2, I think you do a good job presenting the findings, but perhaps you could discuss the claim that the media “naturalizing official sources’ narratives, rather than being sceptical or holding them accountable. I’m not convinced your analysis shows this process of naturalizing. It would be helpful if you could explain and outline more specifically what thesis and possible antithesis you identified as well as the narrative, and how this shifted. I don’t understand why you speculate about the audience’s trust in the media coverage as there is nothing empirical data in our analysis about trust. The last argument in this section – the “new” role of mediators for powers”- needs more elaboration, in relation to your discussion with the literature review and the empirical data. 

Author Response

Reviewer 1

Thank you for the reviewed article. Overall, I think it has improved after the first round of reviews and the below suggestions for minor changes are associated with:

-          1) Interpretation and conclusions – your claim about the new role of the media as mediators for power - is warranted by and sufficiently derived from/focused on the data.

-          2) Key messages/What this paper adds – reflect accurately what the article says and what topics you can’t say something about (for instance trust). Practical implications?

-          3) discourse analysis – more specific about the analytical design and the discursive construction/ narratives

Thank you for your feedback. We are glad that the paper has improved. We have made further changes to the paper in response to the additional feedback, and we hope that the paper is now ready for publication.

Abstract

I really value how you point out key findings (the overuse of items implying certainty and the changing roles of the watchdog to the mediator).  However, I am not sure you are entitled to claim that this decrease “the public’s trust in the accuracy of the Covid-19 information presented in the media”. This is mere speculation by the authors.

Thank you for this comment. We have added references and data in the main text (on page 9) that suggest that public trust in the media has indeed decreased in Austria and Germany during the pandemic. This is thus not mere speculation. We have not made additional changes to the abstract, also because our use of “arguably” already mitigates the claim.

Part 1 Introduction

This section has a better focus now, but the argument based on Wegwarth et al., 2022 that policymakers and health experts sometimes shy away from communicating scientific uncertainty needs more support and/or discussion. Research from my own country suggests the opposite,

-          Kjeldsen, J. E., Mølster, R., & Ihlen, Ø. (2022). Expert uncertainty: Arguments bolstering the ethos of expertise in situations of uncertainty. In The pandemic of argumentation (pp. 85-103). Cham: Springer International Publishing.

-          Ihlen, Ø., Just, S. N., Kjeldsen, J. E., Mølster, R., Offerdal, T. S., Rasmussen, J., & Skogerbø, E. (2022). Transparency beyond information disclosure: strategies of the Scandinavian public health authorities during the COVID-19 pandemic. Journal of Risk Research25(10), 1176-1189.

In these studies, the authors argue that health experts acknowledge uncertainty to bolster their credibility, and as such, may not have negative effects on people’s trust in the communicators. The studies also provide you with studies backing your argument that such communication can be risky.

We have added brief information about the Scandinavian case to the main text, and we now cite both of the publications given above.

Moreover, I don’t understand the argument that “it stands to reason that newspapers no longer see themselves as a corrective will uncritically echo….” P. 2 (there is also a “that” too much). I think this is a harsh comment about the professionalism of journalism. To my knowledge, journalists worldwide take the challenges associated with fake news, fact-checking, propaganda etc seriously. Perhaps you could add some studies supporting your argument of the “selective echoing press”.

We did not find a redundant “that”. In the above quote, it should be “it stands to reason that newspapers that no longer see themselves as a corrective will uncritically echo….”, with two instances of “that”. We have also made the comment less harsh by replacing “will” with “are likely to”, so that we now say that if newspapers no longer see themselves as a corrective, then they will be “likely to” uncritically echo official notions of certainty. 

Part 1.1 Media, sources, and the use of certainty

This section has also improved. I think it was the right decision to delete chunks of text not directly relevant to the analysis. Perhaps you could give more space to the notion of the accountability role of the news media, and whether this is a more important function during the pandemic than under normal situations.  

We have added another sentence to the paragraph that addresses the role of the media in times of crisis.

Part 2 Methods

Please elaborate on what kind of “discourse analytic approach” you are using and how this relates to the very uncertain context of the pandemic.

We have added additional information about the kind of discourse analytic approach. Specifically, we have added the following information in section 1.4: “We employed concepts from Critical Discourse Analysis (Fairclough, 2015) that we deemed relevant for the goals of the analysis, with a focus on modality and generic references.” Additionally, we have clarified the following sentence in section 2.4: “The scale of epistemic modality reaches from low probability (may/perhaps) to high likelihood (must/definitely), with the highest probability being expressed by the complete absence of modality, i.e., by simply stating something as a fact, for example, by using the generic present tense and/or a generic referent, as in Science agrees….”

Part 2.4 Analyzing certainty

This is a good paragraph. It is important to define this very relevant term. It would be useful to highlight the difference between expert alternative voices outside the authoritative governmental sphere, as you later claim that the media does not give space to such voices outside the “sanctioned spectrum”.

It is not entirely clear to us what highlighting “the difference between expert alternative voices outside the authoritative governmental sphere” means. What kind of difference is the reviewer referring to? Since we focus on certainty in the paper (implemented as a focus on modality and generic references), we can’t really compare certainty in those alternative voices, since these alternative voices had too small of a presence in the newspapers we selected.

Part 3 results

Reducing the descriptive text has made this section more interesting and to the point. I’m not convinced about the discourse analytic approach. Which discourses have you identified? What are their functions?

As we have now clarified in the methods section, our discourse analytic approach has focused on the aspects of modality and generic reference because these are linguistic markers related to certainty. We hope this is now sufficiently explained in the methods section.

Please elaborate on and show how exactly “the corpus data suggests that indeed the media have closely followed, observed and echoed the official narrative of the pandemic, rather than critically questioning or scrutinizing it when this was warranted”.

We present the official narrative for herd immunity at the beginning of section 3.1 and the official narrative for Vaccine Efficacy at the beginning of section 3.2. For clarification, we have expanded the information about the official narrative for herd immunity.

Also, you need to be specific about the findings of “the changing roles of the watchdog to the mediator” – how and when did this change occur?

The goal of our analysis is to see to what extent the media have echoed the official narrative of the covid response. Our goal is not to pinpoint how and when this change occurred, which would require a larger scale diachronic study, which is outside the scope of this paper.

Part 4 Discussion and conclusion

I would very much see a closer discussion of the research question in relation to the literature review on both media’s role, sources, and the use of scientific information. Sometimes I get a feeling that you are shooting the messenger and accusing the media of misinforming the audience purposely. The text comes across as speculative.

We believe that the media have a responsibility to vet the sources and information they choose to deliver to the public and to present a diverse range of, in this case scientific, perspectives to best inform their audience. Therefore, while they are the messenger and are not responsible for the content promoted by their sources, they are responsible for putting the sources’ claims into a broader perspective.

Perhaps it is more fruitful to discuss the media’s use of sources and not the vaccination program per se.

Thank you for the suggestion of looking at the media’s use of sources. This is a really interesting topic and one that we might look at in future work, but it is beyond the scope of this paper.

I relation to RQ2, I think you do a good job presenting the findings, but perhaps you could discuss the claim that the media “naturalizing official sources’ narratives, rather than being sceptical or holding them accountable. I’m not convinced your analysis shows this process of naturalizing. It would be helpful if you could explain and outline more specifically what thesis and possible antithesis you identified as well as the narrative, and how this shifted.

We have now added more information about the official narrative to section 3.1., and we hope it is now clearer how our analysis suggests that the media were mostly echoing official sources rather than being skeptical. They do that by employing an unwarranted amount of certainty in reporting their sources’ claims.

I don’t understand why you speculate about the audience’s trust in the media coverage as there is nothing empirical data in our analysis about trust.

Thank you for pointing this out. We had previously only cited the empirical data about the audience’s trust in the media coverage, and we now provide the actual figures that suggest that trust in the media has declined during the pandemic in Austria (and in Germany). Specifically, we now write “This becomes an issue of interest if we consider that words like efficacy are likely to raise expectations in news readers and, when those are not fulfilled, this might lead to media consumers being more critical of media and their coverage of scientific issues. In line with this suggestion, public trust in the media in Austria has decreased by 11% from mid-2021 to mid-2022 (OGM, 2022) and a study from Germany suggests that 41% of people think that the credibility of journalism has declined as a result of Corona reporting (TU Dortmund, 2022).”

The last argument in this section – the “new” role of mediators for powers”- needs more elaboration, in relation to your discussion with the literature review and the empirical data. 

Throughout the paper, we argue that the frequent use of statements implying certainty with regards to the official narrative (as suggested by our qualitative analysis) is “the clearest indicator of this new role of the media – as mediators for power, rather than as a counterweight to power”. While our final sentence epitomizes this claim, we believe the argument runs consistently through the entire paper.
